# Pressure-induced transition from a Mott insulator to a ferromagnetic Weyl metal in La$_2$O$_3$Fe$_2$Se$_2$

Ye Yang[1,5], Fanghang Yu[1,5], Xikai Wen[1], Zhigang Gui[1], Yuqing Zhang[1], Fangyang Zhan[2], Rui Wang [2✉], Jianjun Ying [1✉] & Xianhui Chen [1,3,4✉]

The insulator-metal transition in Mott insulators, known as the Mott transition, is usually accompanied with various novel quantum phenomena, such as unconventional superconductivity, non-Fermi liquid behavior and colossal magnetoresistance. Here, based on high-pressure electrical transport and XRD measurements, and first-principles calculations, we find that a unique pressure-induced Mott transition from an antiferromagnetic Mott insulator to a ferromagnetic Weyl metal in the iron oxychalcogenide La$_2$O$_3$Fe$_2$Se$_2$ occurs around 37 GPa without structural phase transition. Our theoretical calculations reveal that such an insulator-metal transition is mainly due to the enlarged bandwidth and diminishing of electron correlation at high pressure, fitting well with the experimental data. Moreover, the high-pressure ferromagnetic Weyl metallic phase possesses attractive electronic band structures with six pairs of Weyl points close to the Fermi level, and its topological property can be easily manipulated by the magnetic field. The emergence of Weyl fermions in La$_2$O$_3$Fe$_2$Se$_2$ at high pressure may bridge the gap between nontrivial band topology and Mott insulating states. Our findings not only realize ferromagnetic Weyl fermions associated with the Mott transition, but also suggest pressure as an effective controlling parameter to tune the emergent phenomena in correlated electron systems.

Mott-insulating behavior is induced by strong electron correlations, which are considered to associate with many exotic quantum phenomena of matter such as unconventional superconductivity and quantum spin liquids[1,2]. The Mott-insulating state is present when the repulsive Coulomb potential $U$ is large enough to open an energy gap. By controlling band filling, $U$, and bandwidth, the Mott transition would occur once the Mott-insulating state tunes to the metallic state, in which unconventional superconductivity is often expected[3]. As an effective controlling parameter, pressure is often used to tune the Mott insulators, and is thought as a useful method to find unconventional superconductors. Indeed, superconducting states emerging incidental to the pressure-induced Mott transition was studied in some systems[4–8]. However, to date, such cases have been still only limited to a few materials. Beyond superconducting states, other exotic quantum states associated with the Mott transition may also take place[9–15], which have been largely unexplored. Especially, as nontrivial band topology is the

[1]Department of Physics, and CAS Key Laboratory of Strongly-coupled Quantum Matter Physics, University of Science and Technology of China, Hefei, Anhui 230026, China. [2]Department of physics & Center of Quantum Materials and Devices & Chongqing Key Laboratory for Strongly Coupled Physics, Chongqing University, Chongqing 400044, China. [3]CAS Center for Excellence in Quantum Information and Quantum Physics, Hefei, Anhui 230026, China. [4]Collaborative Innovation Center of Advanced Microstructures, Nanjing University, Nanjing 210093, China. [5]These authors contributed equally: Ye Yang, Fanghang Yu. ✉e-mail: rcwang@cqu.edu.cn; yingjj@ustc.edu.cn; chenxh@ustc.edu.cn

most interesting feature in realistic materials recently[16,17], the exploration of the relationship between nontrivial band topology and Mott transition is highly desirable.

To search for more abundant exotic quantum states associated with the Mott transition, we focused on an iron oxychalcogenide compound $La_2O_3Fe_2Se_2$, a well-known Mott insulator. The structure of $La_2O_3Fe_2Se_2$ has a close relationship with the Fe-based high-Tc super-conductor, and the 3d electrons of iron play a key role in the physical properties of both materials[18–29]. The $Fe_2O$ transition-metal layers are a rare example of the anti-$CuO_2$ type and can also be described as a network of face-sharing octahedra, where the transition-metal-centered octahedron is made up of two axial oxide ions and four equatorial selenide ions. This material, and its oxysulfide analog, have semiconducting properties and have been considered as correlation-induced Mott insulators with the antiferromagnetic (AFM) ground state[21,25]. While exhibiting strongly correlated Mott-insulating behavior, $La_2O_3Fe_2Se_2$ may offer tunability of their electronic properties near a metal-insulator transition. Here, by applying the pressure, we were able to tune the Mott-insulating state to ferromagnetic (FM) metallic state, which may prevent the emergence of superconductivity in $La_2O_3Fe_2Se_2$. The pressure-induced insulator-metal transition was also revealed by our theoretical calculations. We found that the pressure-driven transition from the AFM insulating phase to FM metallic phase is always present when considering the different $U$ values, which only affect the transition pressure of Mott transition. With an appropriate $U$ value, the calculated transition pressure of Mott transition agrees well with the experimental value ~37 GPa, which verifies the enlarged bandwidth and shrinkage of $U$ at high pressure. Furthermore, based on symmetry analysis and parity argument, we uncover that the high-pressure FM metallic phase possesses nontrivial

band topology and its topological properties can be easily tuned by the magnetic field. With the magnetization direction along the easy axis, six pairs of Weyl points (WPs) in the vicinity of Fermi level are present, and two pairs of them located on the high-symmetry lines are protected by the twofold rotation symmetry. These discoveries suggest high pressure as an effective method to tune the correlated electron materials and search for promising topological states.

## Results and discussion

### Pressure-induced Mott transition in $La_2O_3Fe_2Se_2$

The crystal structure of $La_2O_3Fe_2Se_2$ is presented in Fig. 1a. There are two typical layered units of $La_2O_2$ and $Fe_2OSe_2$, which stack along the crystallographic $c$ axis. Figure 1b displays the x-ray diffraction pattern of a $La_2O_3Fe_2Se_2$ single crystal. Only (00 l) diffraction peaks can be detected, indicating the pure phase of the as-grown single crystal with a [001] preferred orientation. The $c$ axis lattice parameter is determined to be 18.622 Å, which is consistent with the previously reported value of $La_2O_3Fe_2Se_2$[18]. We performed the high-pressure resistivity measurements on the $La_2O_3Fe_2Se_2$ single crystal as shown in Fig. 1c, d. The resistivity shows insulating behavior at low pressure consistent with the ambient-pressure measurement[25]. By increasing the pressure, the resistivity can be gradually suppressed. With pressure above 37 GPa, the resistivity shows metallic behavior evidencing for a pressure-induced insulator-metal transition. In the metallic state, we can see a weak kink around 120-140 K from the resistivity curves as shown in Fig. 1d, which possibly is due to the magnetic phase transition at low temperature. The transition temperature can be determined as the arrows indicated in Fig. 1d. The weak upturn of the resistivity at low-temperature can possibly be attributed to the Kondo effect[30], since the low-temperature resistivity shows a logarithmic increase and

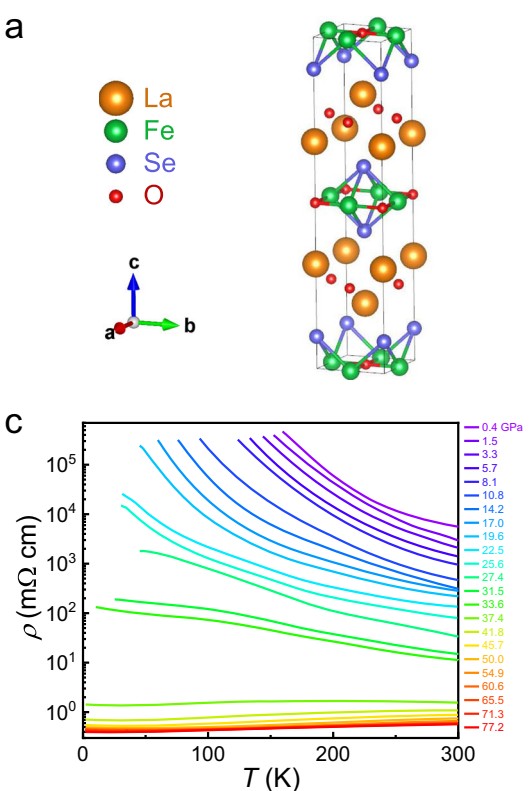

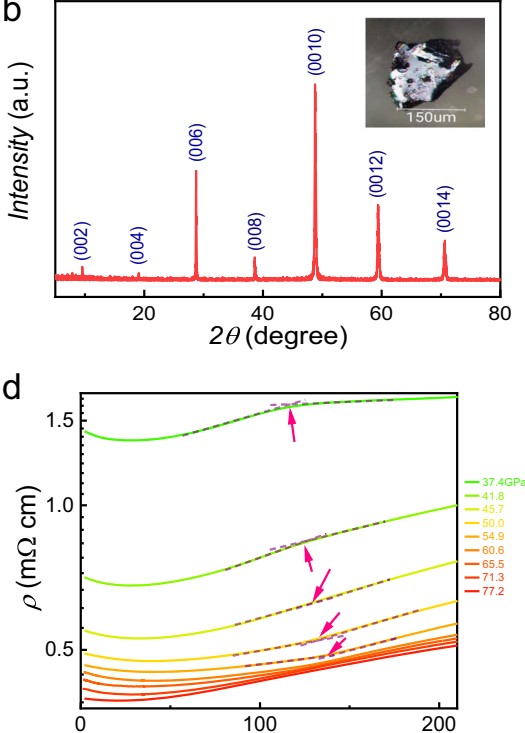

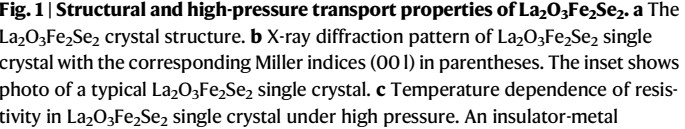

**Fig. 1 | Structural and high-pressure transport properties of $La_2O_3Fe_2Se_2$. a** The $La_2O_3Fe_2Se_2$ crystal structure. **b** X-ray diffraction pattern of $La_2O_3Fe_2Se_2$ single crystal with the corresponding Miller indices (00 l) in parentheses. The inset shows photo of a typical $La_2O_3Fe_2Se_2$ single crystal. **c** Temperature dependence of resistivity in $La_2O_3Fe_2Se_2$ single crystal under high pressure. An insulator-metal

transition occurs near 40 GPa. **d** Enlarged area of the temperature dependence of resistivity for $La_2O_3Fe_2Se_2$ single crystal when entering the metallic state. The pink arrows indicate the kinks in the resistivity curves which is possibly related to the ferromagnetic transition.

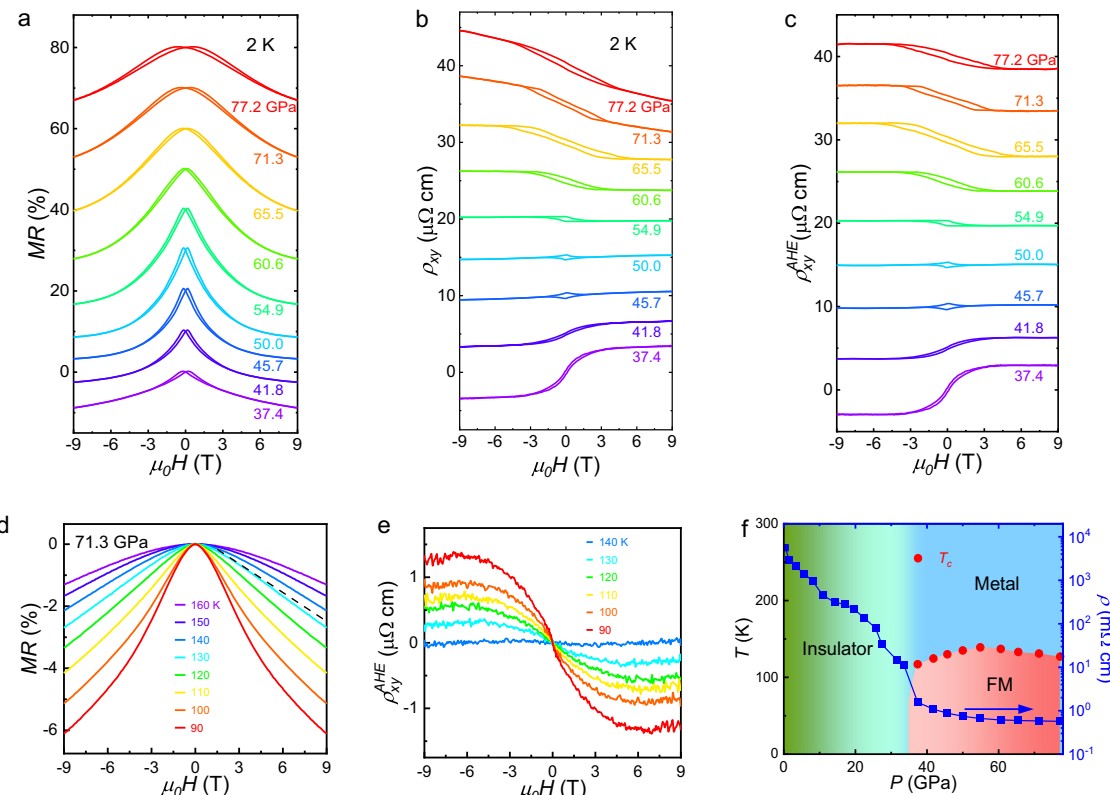

**Fig. 2 | High-pressure phase diagram of La$_2$O$_3$Fe$_2$Se$_2$. a** Pressure dependence of the magnetoresistance for pressurized La$_2$O$_3$Fe$_2$Se$_2$ single crystal measured at 2 K. **b** The Hall resistivity measured at 2 K under various pressure. **c** The extracted anomalous Hall resistivity under various pressures. The MR **d** and Hall coefficient **e** measurements at various temperatures with pressure at 71.3 GPa. **f** Pressure-temperature phase diagram of La$_2$O$_3$Fe$_2$Se$_2$ derived from the electric transport measurements.

gradually becomes nearly saturated toward low temperatures as shown in Supplementary Fig. 2.

In order to figure out what happens below the transition temperature, we perform the magnetoresistance (MR) and Hall effect measurements with magnetic field applied along $c$ axis direction under various pressures at 2 K as shown in Fig. 2a, b, respectively. All the MR curves show a butterfly behavior, indicating it entered the FM ground state at low-temperature for compressed La$_2$O$_3$Fe$_2$Se$_2$. Such FM ground state is also confirmed from the Hall effect measurement. The Hall resistivity shows hysteresis behavior, consistent with the MR measurement. We can also extract the anomalous Hall resistivity $\rho_{xy}^{AHE}$ by subtracting the high-field linear ordinary Hall term as shown in Fig. 2c. We can clearly see that $\rho_{xy}^{AHE}$ changes sign around 50-55 GPa, at the meantime, the ordinary Hall also changes sign and the magnitude of MR reaches the maximum value as shown in Supplementary Fig. 3. Such behavior can be possibly attributed to the pressure-induced Lifshitz transition and variation of berry curvature under pressure. We also notice that the sharpness of the hysteresis curve and the coercive field are different at different pressures, which can be related to the change of the FM grain size and interaction between different FM grains at high pressure. To confirm the resistivity anomaly as shown in Fig. 1d is associated with the FM transition, we also perform the MR and Hall coefficient measurements at various temperatures with pressure at 71.3 GPa as shown in Fig. 2d, e, respectively. The $\rho_{xy}^{AHE}$ gradually suppressing with temperature increasing until completely suppressed above 130 K, consistent with Curie temperature (T$_C$) determined from the R-T measurements as shown in Fig. 1d. At the meantime, the MR shows U shape behavior above 130 K as shown in Fig. 2e, indicating the T$_C$ is around 130 K, consistent with the Hall resistivity measurement. We can further determine the spin orientation of the FM state at high

pressure from the anisotropy MR results in Supplementary Fig. 4. The MR measurement shows hysteresis behavior below 4 T with magnetic along $c$ axis direction. While this behavior occurs when the magnetic field is only below ~2 T along ab plane, indicating the spin orientation tend to the ab plane in the FM state. Moderate magnetic field can fully polarize the spin, which would dramatically modify the band structure with different spin orientations in this material. We can map out the phase diagram with pressure for La$_2$O$_3$Fe$_2$Se$_2$ in Fig. 2f. The measured resistivity at 300 K exhibits a sudden reduction around 37 GPa, clearly signaling an insulator-metal transition, and a dome-like FM phase emerges at low-temperature in the metallic phase.

To clarify the origin of insulator-metal transition in La$_2$O$_3$Fe$_2$Se$_2$, we performed high-pressure XRD measurements as shown in Fig. 3. We find that the structure does not change with pressure. The XRD patterns collected at different pressures are displayed in Fig. 3a. We performed Le Bail fit for all the XRD patterns, and all the XRD patterns can be well fitted using its ambient-pressure structure. Selected fitting results are shown in Supplementary Fig. 5 in the Supplemental Materials. We notice that some ambient-pressure peaks disappear at high pressure, which originate from either overlapping with the other peaks or the low intensity induced by peak broadening. We can extract the lattice parameters $a$ and $c$ as a function of pressure shown in Fig. 3b. The derived cell volume is shown in Fig. 3c, which can be well fitted by using the Birch-Murnaghan equation of state with the derived bulk modulus $B_0$ = 73.9 GPa. We note that the $c/a$ decreases with increasing pressure, indicating the interlayer coupling becomes more pronounced at high pressure. In addition, the slope of pressure dependence of $c/a$ suddenly changes ~40 GPa, close to the pressure at which insulator-metal transition takes place.

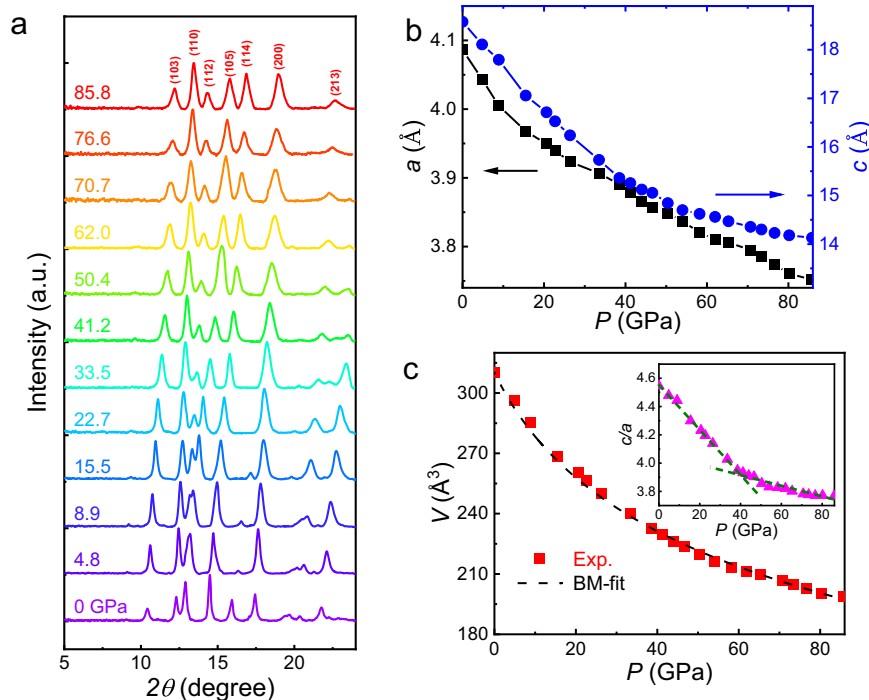

**Fig. 3 | High-pressure XRD of La₂O₃Fe₂Se₂.** **a** XRD patterns of La₂O₃Fe₂Se₂ poly-crystal under high pressure up to 85.8 GPa with an incident wavelength $\lambda = 0.6199$ Å. No structural phase transition is detected. **b** Pressure dependence of the lattice parameters $a$ and $c$ for La₂O₃Fe₂Se₂. **c** The derived cell volume as a function of pressure for La₂O₃Fe₂Se₂. The black dashed line is a fitted curve by the Birch-Murnaghan equation of state with the derived bulk modulus $B_0 = 73.9$ GPa. The inset shows the pressure dependence of the c/a ratio which shows an anomaly ~40 GPa, close to the insulator-metal transition.

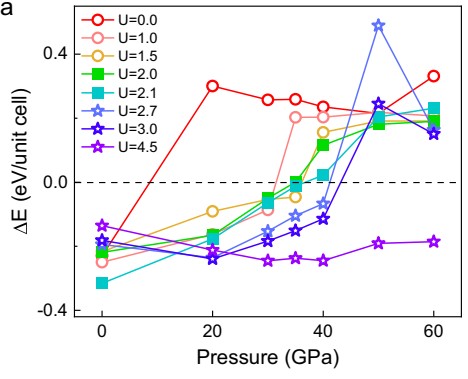

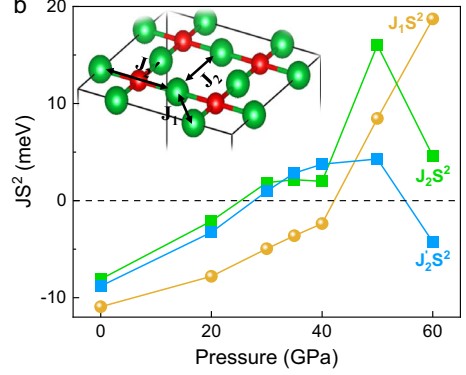

**Fig. 4 | Theoretical investigation of pressure-induced magnetic transition.** **a** magnetic phase transition of La₂O₃Fe₂Se₂ polycrystal within different Hubbard $U$ values. **b** Exchange coupling parameters as a function of pressure within $U = 2.1$ eV. The inset illustrate three exchange interactions, including the nearest-neighbor interaction of Fe-Fe contribution ($J_1$), the next-nearest-neighbor interaction of Fe-Se-Fe contribution ($J_2$), and the next-nearest-neighbor interaction of Fe-O-Fe contribution ($J_2'$).

## First-principles calculations on La₂O₃Fe₂Se₂ at high pressure

Next, we carried out first-principles calculations to reveal the pressure-induced physics in La₂O₃Fe₂Se₂. Considering the correlated effects of Fe 3d electrons, we used the DFT + $U$ scheme in the framework of on-site Hubbard energy $U$ to investigate the insulator-metal transition and electronic band structures at high pressure. As shown in Supplementary Fig. 6, the calculated lattice parameters agree well with the experimental values until the applied pressure is up to 60 GPa. The narrower bands of Fe atoms in La₂O₃Fe₂Se₂ promote strong correlation strength of 3d electrons, thus resulting in the Mott-insulating state[25]. The magnetism of La₂O₃Fe₂Se₂ is dominated by the Fe atoms of Fe₂OSe₂ layer[25,26]. As discussed in the previous studies[19,20,25,26,31], there are several potential magnetic configurations of the Fe₂OSe₂ layer, as included in Supplementary Fig. 7. We here adopt the convention of magnetically ordered states used in the literatures and then investigate

the evolution of magnetically ordered states of La₂O₃Fe₂Se₂ under pressure. At ambient pressure, our calculated results indicate that the magnetic ground state of La₂O₃Fe₂Se₂ belongs to the AFM3 collinear frustrated configuration, which was confirmed by the elastic neutron-scattering measurement[26]. As shown in Fig. 4a, we plot the relative energies ΔE of the FM and AFM3 magnetically ordered states as a function of pressure. To investigate the influence of correlated effects of Fe 3d electrons on the pressure-driven transitions, we calculate ΔE for a series of $U$ values. Here, the ΔE represents the energy difference between AFM3 state and FM state within certain $U$ value. When the ΔE is positive value, the system would transition to the AFM3 state. One can find that different values of $U$ only affect the transition pressure, and the pressure-induced magnetic transition from the initial AFM state to FM state can always occur with the increase of pressure. When the value of $U$ is ~2.1 eV, the transition pressure is ~40 GPa, which is

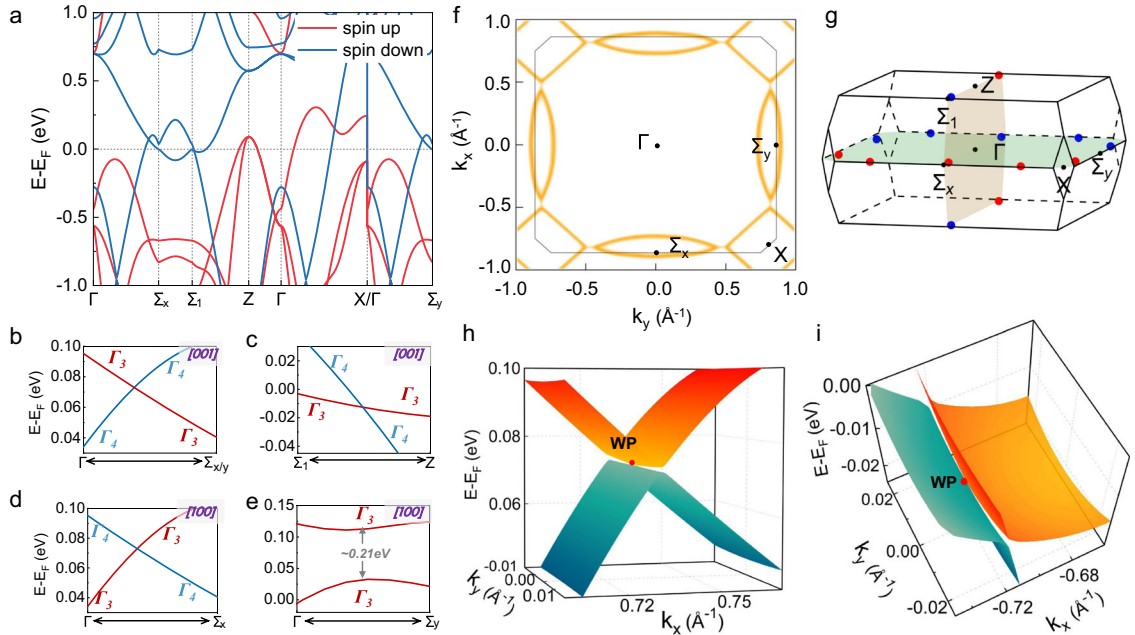

**Fig. 5 | The electronic structure of La₂O₃Fe₂Se₂ under 48 GPa. a** For $U$ value is set to 2.1 eV, Calculated band structure of La₂O₃Fe₂Se₂ without SOC under 48 GPa. **b**–**e** are Zoom-ins of band structures in the present of SOC along [001] and [100] magnetization, respectively. **f** The nodal lines diagram of the $k_x$-$k_y$ plane with $k_z = 0$.

**g** The six pairs of Weyl points survive in the BZ when the applied magnetic field along [100] direction. **h, i** show the type-I Weyl point in the Γ–Σ$_x$ (or Γ–Σ$_y$) path and the type-II Weyl point in Σ$_1$-Z path, respectively.

consistent with our resistivity measurement. The case of $U$ ~2.1 eV is typically smaller than that in AFM Mott-insulating phase in ambient condition[25]. The decrease of electron correlation at high pressure indicates the realization of Mott transition. Besides, we also need to emphasize that it is difficult to determine the accurate values of $U$ at different pressures.

The magnetic ground state is dominated by the spin interactions of magnetic atoms. For the Mott-insulating La₂O₃Fe₂Se₂, its spin Heisenberg Hamiltonian can be modeled using three principal effective parameters $J_1$, $J_2$, and $J_2'$, which respectively represent the nearest-neighbor interaction of Fe-Fe contribution, the next-nearest-neighbor interaction of Fe-Se-Fe contribution, and the next-nearest-neighbor interaction of Fe-O-Fe contribution [see Fig. 4b][19,25,26]. Therefore, to capture the natures of the pressure-induced magnetic transition in La₂O₃Fe₂Se₂, we calculate the interaction parameters $J_1$, $J_2$, and $J_2'$ as a function of pressure, as shown in Fig. 4b. It is clear to see that all three parameters undergo a sudden change around 40 GPa. Especially, $J_1$ quickly varies from negative to positive once the pressure exceeds the critical value. This means a transition of magnetic ground state from AFM Mott-insulating phase to FM metallic phase[32,33]. Our analysis is in good agreement with spin interactions strongly dependent on bandwidth of magnetic atoms.

We further depict the electronic properties of La₂O₃Fe₂Se₂ in FM metallic phase after the insulator-metal transition. We employed DFT + $U$ calculations with $U = 2.1$ eV to investigate the band topology of La₂O₃Fe₂Se₂ under 48 GPa, at which all the WPs would be close to the Fermi level. In the spin-polarized calculations, without spin-orbital coupling (SOC), the electronic band structures along the high-symmetry lines at a pressure of 48 GPa are plotted in Fig. 5a. It is visible that the internal exchange field of FM-ordered states lift the spin degeneracy, resulting in two spin states present different electronic properties around the Fermi level. We can see the majority spin bands show crossing points in the Γ–Σ$_{x/y}$ and Z–Σ$_1$ directions at ~0.07 eV above and ~−0.01 eV below the Fermi level, forming holes and electron pockets. These crossings are in the $k_x$–$k_y$ plane with $k_z = 0$ or $k_z = \pi/c$, respecting to mirror-reflection symmetry $M_z$. We check the irreducible

representations (IRs) of bands and find that the crossing bands belong to opposite mirror eigenvalues ±1 of $M_z$, which give rise to mirror symmetry-protected nodal lines. The electronic properties in the $k_z = 0$ and $k_z = \pi/c$ planes are similar, and thus we mainly focus on the cases of $k_z = 0$ in the following. In Fig. 5f, we show the band-gap between the valence band and conduction band in the $k_x$–$k_y$ plane with $k_z = 0$. From the features of zero band-gap, it is clearly to see that there are three closed nodal lines related to the four-fold rotation symmetry $C_{4z}$ around the $k_z$ axis. One is enclosed to X point, and the other two are enclosed to Σ$_{x/y}$ points, respectively.

In the absence of SOC, the spin and orbitals are regarded as different subspaces, so two spin states are decoupled and each one possesses all symmetry group elements of La₂O₃Fe₂Se₂. After introducing SOC, two spin states couple together, which will generally lower symmetries of crystals. The group elements of corresponding magnetic space group are dependent on the magnetization direction. For instance, as the magnetization direction is along the [001] direction (i.e., the $z$ axis), the symmetry of La₂O₃Fe₂Se₂ is reduced to the magnetic point group $C_{4h}$. In this case, the mirror-reflection symmetry $M_z$ is reserved. As shown in Fig. 5b, c, the crossing bands along the Γ–Σ$_x$ and Γ–Σ$_y$ directions belong to different IRs Γ₃ or Γ₄ of little group $C_s$, indicating that the nodal lines in the $k_x$–$k_y$ plane with $k_z = 0$ (or $k_z = \pi/c$) are indeed preserved. Once the magnetization direction is deviated from the [001] direction, the mirror-reflection symmetry $M_z$ is broken, leading to the nodal lines in the reflection-invariant plane would vanish.

To elucidate the magnetization-dependent topological features of La₂O₃Fe₂Se₂, we perform total energy calculations to determine the magnetization directions. The results show the energetically most favorable magnetization direction isotopically lies in the $a$–$b$ plane (i.e., the $x$-$y$ plane) of La₂O₃Fe₂Se₂, consistent with our MR measurement. Without loss of generality, we consider the case of the [100] magnetization since the analyses for other magnetization directions are essentially similar. When the magnetization is along the [100] direction (i.e., the $x$ axis), the magnetic point group of La₂O₃Fe₂Se₂ is reduced to $C_{2h}$, which contains typical group elements as inversion $I$,

twofold rotation along the [100] direction $C_{2x}$, the product $TC_{2z}$ of time-reversal $T$ and twofold rotation along the [001] direction $C_{2z}$. The antiunitary symmetry $TC_{2z}$ forbids the presence of nodal lines in the $k_x–k_y$ plane with $k_z = 0$ (or $k_z = \pi/c$) but allow the existence of WPs[34]. As shown in Fig. 5d, e, we can see that the band crossings are preserved and gapped along the $\Gamma–\Sigma_x$ and $\Gamma–\Sigma_y$ directions, respectively. Through carefully search the local minimum of band structures, we find that there are six pairs of WPs, which are symmetrically distributed in momentum space and related to the inversion symmetry $I$ (see Fig. 5g). The detailed information of WPs, such as coordinates, chirality, and energies related to Fermi level $E_F$ are concluded in Supplementary Table 2. It is worth noting that the WPs along the $k_x$ axis (i.e., the $\Gamma–\Sigma_x$ or $Z–\Sigma_1$) are protected by the $C_{2x}$ rotation; that is, their crossing bands belong to different eigenvalues ±1 of $C_{2x}$ [see Fig. 5d]. The other WPs are accidental nodal points. Interestingly, although the WPs along $\Gamma–\Sigma_x$ or $Z–\Sigma_1$ are associated with the same little group $C_{2x}$, they belong to different types of WPs. As shown in Fig. 5h, i, we can see that the WP along $\Gamma–\Sigma_x$ is type-I and WP along $Z–\Sigma_1$ is type-II. To understand the relationship between band topology and the emergence of WPs, we also have done a parity analysis of the occupied Bloch states at the eight time-reversal invariant momenta (TRIM) points. The results show that the product of the occupied bands running over all TRIM points is 1 but there are two parallel time-reversal invariant planes (i.e., $k_2 = 0$ and $k_2 = 0.5$) with nonzero Chern number $C = 1$ [see Supplementary Fig. 8 in the Supplementary information]. In this case, the gapless states in a semimetal must require the presence of 2D planes with $C = 0|_{k_2 = k_i} (0 < k_i < 0.5)$, resulting in even pairs of WPs in the Brillouin zone[35]. In FM phase of $La_2O_3Fe_2Se_2$, the presence of six pairs of WPs indicates that there are two 2D planes with $C = 0|_{k_2 = k_i} (0 < k_i < 0.5)$[35,36], and thus the parity argument further confirms the nontrivial band topology of $La_2O_3Fe_2Se_2$ at high pressure. Moreover, we further calculate the anomalous Hall conductivity (AHC) $\sigma_{xy}$ when the magnetic field is applied along the [001] direction. As shown in Supplementary Fig. 9, the intrinsic AHC would change sign since the pressure could impact the relative position of chemical potential, which agrees with our experimental observation.

In conclusion, by combining the high-pressure transport and XRD measurements, as well as the first-principles calculations and symmetry analysis, we find a rare example, that is, the iron oxychalcogenide $La_2O_3Fe_2Se_2$ can be tuned from Mott insulator to FM Weyl metal at high pressure without structural phase transition. Our work demonstrates high pressure as an effective method to tune the ground state in strongly correlated materials and search for exotic topological quantum states. More importantly, these findings may facilitate the understanding of the relationship between nontrivial band topology and Mott transition.

## Methods

### Sample growth

Polycrystalline sample of $La_2O_3Fe_2Se_2$ was synthesized by solid-state reaction method using $La_2O_3$, Fe powder, and Se powder as starting materials. $La_2O_3$ was dried by heating in air at 1000 °C for 12 h before using. The raw materials and precursors were weighed according to the stoichiometric ratio, thoroughly grounded, pressed into pellets, and then sealed in evacuated quartz tubes. The sealed tubes were heated to 1000 °C and held for 2 days. In order to improve their purity and homogeneity the resulting products were reground in argon atmosphere before applying a second heat treatment at the same temperature. The polycrystal $La_2O_3Fe_2Se_2$ were loaded into an alumina crucible and then sealed in an iron crucible. The iron crucible was heated to 1300 °C and kept for 12 h in vacuum. Subsequently, the temperature was lowered slowly to 1000 °C in 10 days, and then the iron crucible was cooled in the furnace by shutting off the power. Plate-like single crystals with typical size ~100 μm can be yielded at the bottom of the alumina crucible. A representative piece of single crystal with thickness ~10 μm is shown in the inset of Fig. 1b. Single crystal X-ray diffraction data were collected at room temperature using an X-ray diffractometer (SmartLab-9, Rikagu) with Cu Kα radiation and a fixed graphite monochromator. Single crystals were used for the high-pressure transport measurements. Several pieces of single crystals were crushed to powder and used for high-pressure XRD experiments.

### High-pressure experiments

Diamond anvils with 200 μm culet size and c-BN + epoxy gasket with a sample chamber diameter of 75 μm were used for high-pressure transport measurements. One piece of $La_2O_3Fe_2Se_2$ single crystal was cut to the dimensions of 50 μm × 50 μm × 10 μm and loaded into the sample chamber. NaCl was used as a pressure transmitting medium and the pressure was calibrated by using the shift of ruby florescence and diamond anvil Raman at room temperature. For each measurement cycle, the pressure was applied at room temperature using the miniature diamond anvil cell[37]. The transport measurements were carried out using the Quantum Design PPMS9 or cryostat (HelioxVT, Oxford Instruments).

The high-pressure synchrotron XRD measurements were performed at room temperature at the beamline BL15U1 of the Shanghai Synchrotron Radiation Facility (SSRF) with a wavelength of $\lambda = 0.6199$ Å. A symmetric diamond anvil cell with a pair of 200 μm culet size anvils was used to generate pressure. A 70 μm sample chamber was drilled from the Re gasket and Daphne 7373 oil was loaded as a pressure-transmitting medium.

### First-principles calculations

We carried out the first-principles calculations based on the density-functional theory (DFT)[38,39] as implemented in Vienna ab initio simulation package[40]. The electron-ion interaction was treated by the projector-augmented-wave method[41]. A plane wave expansion with an energy cutoff of 500 eV was used. The generalized gradient approximation (GGA) with Perdew–Burke–Ernzerhof[42] functional was chosen to describe the exchange-correlation interactions. The ionic relaxation and electronic self-consistent iteration were converged to 0.001 eV/Å³ and $10^{-7}$ eV. To study the pressure-induced magnetic transition, we build 2 × 2 supercell for all selected magnetic structures. The relative energy ΔE can describe by $\Delta E = 0.25 \times (E_{AFM} – E_{FM})$, in which the ΔE represents the energy difference between AFM state and FM state; $E_{AFM}$ is the total energy of relaxed structure with AFM state; $E_{FM}$ is the total energy of relaxed structure with FM state. When the ΔE is positive value, the system would belong to the AFM state. The irreducible representations (IRs) of little group in momentum space were calculated by using the IRVSP package[43]. To further determine topological properties, we constructed a Wannier tight-binding Hamiltonian by projecting from the Bloch states into the basis of maximally localized Wannier functions in the WANNIER90 package[44]. We use the Wilson-loop method based on the evolution of the average position of Wannier centers[45,46] to calculate the chirality of these nodal points, and find they are all WPs. Then, the topological features such as nodal lines and chirality of WPs were determined using WANNIERTOOLS code[47].

## Data availability

All data supporting the findings of this study are either provided with this article or available from the corresponding authors upon reasonable request. Source data are provided with this paper.

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

## Acknowledgements

This work was supported by the National Key Research and Development Program of the Ministry of Science and Technology of China (Grants no. 2019YFA0704900), the National Natural Science Foundation of China (Grants no. 1188810, no. 11534010, and no. 12222402), the Anhui Initiative in Quantum Information Technologies (Grant no. AHY160000), the Science Challenge Project of China (Grant no.

TZ2016004), the Innovation Program for Quantum Science and Technology (Grant no. 2021ZD0302800), CAS Project for Young Scientists in Basic Research (Grant no.YBR-048), the Key Research Program of Frontier Sciences, CAS, China (Grant no. QYZDYSSWSLH021), the Strategic Priority Research Program of the Chinese Academy of Sciences (Grant no. XDB25000000), the Collaborative Innovation Program of Hefei Science Center, CAS, (Grant no. 2020HSC-CIP014) and the Fundamental Research Funds for the Central Universities (Grants no. WK3510000011). The DFT calculations in this work are supported by the Beijing Super Cloud Computing Center (BSCC) and the Super-computing Center of the University of Science and Technology of China. High-pressure synchrotron XRD work was performed at the BL15U1 beamline, Shanghai Synchrotron Radiation Facility (SSRF) in China.

## Author contributions

X.H.C., J.J.Y., and R.W. conceived the project and supervised the overall research. F.Y. grew the single crystal samples. F.Y., X.K.W., Z.G.G., Y.Q.Z., and J.J.Y. performed the high-pressure measurements. Y.Y., F.Y.Z., and R.W. performed the DFT calculations. Y.Y., J.J.Y., R.W., and X.H.C. wrote the manuscript with input from all authors.

## Competing interests

The authors declare no competing interests.
