## [Peer Review File · Nature Communications]

REVIEWER COMMENTS

Reviewer #1 (Remarks to the Author):

The authors claim that a Weyl ferromagnetic phase emerges under high pressure. The manuscript appears to be a joint experimental + theoretical paper. The authors present a very interesting claim of an AFM to FM phase transition under pressure and this seems to be the most significant and compelling part of the paper. Predictions of a Weyl metallic state are given on the basis of theory claims.

In my view, all of the items that I have labeled with a "*" should be addressed. I feel that addressing these comments is mandatory requirement before publication.

1) * $\text{La}_2\text{O}_3\text{Fe}_2\text{Se}_2$ is not easy to grow. To my knowledge, the single crystal version has not been reported. Since 2010, expert materials growth research groups have found this material extremely difficult to produce in a single crystal format. Perhaps the authors are aware of a report on the successful growth of the single-crystal form of this material. If so, they should include a citation.

a. *The authors should make it clear/easy for the reader to determine whether the paper is making claims about powder samples or single-crystals. The reader should not have to work hard to determine what sample format is being reported. In my opinion the authors should fix this.

b. *The authors need to explicitly mention how they have managed to grow single-crystals of this material. More explicit growth methods are needed.

c. *If the authors are reporting any results about single-crystal samples, what are the sizes of such single-crystal samples used in each experimental method? Otherwise, it should be made more clear that they are conducting experiments on powders.

d. *A picture of a single-crystal is shown in Fig. 1b. Is this a representative size of the single crystals grown/observed in this material? What is the thickness of a piece such as the one shown in Fig 1b? There is no mention of single crystal refinement.

e. Perhaps I missed this point, but why are single-crystals even mentioned in this paper? Do the authors think that their single-crystals might be used to demonstrate presence of WPs if high-pressure ARPES is performed on these materials?

2) *Regarding the powder diffraction data, there are some 0 GPa peaks that do not appear at higher pressure. The authors should address this.

3) *The authors need to give an indication of what the sample quality is for the reported materials. More information on the quality of the Rietveld fit should be given; what are some figures of merit

for the fitting. For example, can the authors provide details on the Rietveld fitting quality parameters, and an indication of their crystals' quality compared to published crystals?

4) *Some details of the diamond anvil cell should be given in the Methods section. If there is a instrumentation paper written about the diamond anvil cell, the authors should cite.

5) *The statement "This implies the existence of six pairs of WPs...." is insufficient. For an article in a general journal, the authors need to provide more explanation of the general readership.

6) *I believe the most important open issue to be understood in this paper is the antiferro-to-ferromagnetic phase transition upon application of external pressure.

7) *The true Mott insulator occurs only in the quantum paramagnetic state. An antiferromagnetic insulator is not really a Mott insulator driven by a competition between hopping and the on-site Coulomb interaction U , but a result of band-folding effects or changes in the unit cell associated with development of antiferromagnetic order.

8) *Have the authors taken into account the effect of U in their ab initio calculations. Nothing seems to be mentioned regarding this aspect of the paper. Is DFT+ U being used or something else? This should be better addressed in the paper.

9) *Thus, the "shrinkage of U at high pressure" (line 63) and the emergence of Weyl fermions remains questionable in my opinion. Are the Weyl points discussed in the paper derived in bare band or in the correlated electron framework? These items are not very clear. A better elucidation should be given.

10) *It is unclear how the authors have calculated " ΔE for a series of U values" (line 146).

11) Can the authors clarify the physical meaning of "Significantly, when the value of U is about 2.1 eV, the critical pressure is around 40 GPa, which satisfies our resistivity measurement."?

12) *Concerning the resistivity measurements, the authors should clarify the origin of the low-temperature resistivity upturn clearly seen in the "metallic phase" at pressure up to 71.3 GPa in Fig. 1d. Is this related to Kondo physics or not? (For details regarding the changes in resistivity due to Kondo effect, see PRB 98, 035114 (2018) and references therein.)

Reviewer #2 (Remarks to the Author):

The authors report a pressure-induced transition from a Mott insulator state to a ferromagnetic Weyl metal state in the iron oxychalcogenide $\text{La}_2\text{O}_3\text{Fe}_2\text{Se}_2$, by combining high-pressure magneto-transport measurements, XRD measurements, and first-principle calculations. I have several questions to the authors as described below;

1. Regarding the resistivity curve under pressure shown in Fig. 1d, why is the kink structure corresponding to the magnetic transition temperature so broad ? Normally, ferromagnetic transition may appear as a clearer kink structure in resistivity. Furthermore, why is the resistivity enhanced at lower temperature (e.g. below ~ 30 K for 37GPa) ? Does this mean that the system is still an insulator at the lowest temperature ?

2. About the Hall resistivity shown in Fig. 2b&c, the sharpness of the hysteresis curve is clearly different below 41.8 GPa and above 45.7 GPa. Also, the coercive field seems to take the minimum around 50 GPa. What are the possible origins of these observations ?

Although the data quality is amazingly high given the very high pressure they have reached, I'm afraid that this work may not attract a broad interest from the readers. One reason is that I cannot see any direct relationship between the Mott physics at the ambient pressure and the band topology at the high-pressure. It is highly natural that a Mott insulator finally becomes a metal under pressure, and the appearance of the Weyl points seems to simply originate from its crystal structure. Another point is that there is no experimental data which is directly related to the presence of Weyl points (e.g. "giant" anomalous Hall effect or chiral anomaly often reported in Weyl semimetals). For these reasons, I recommend its publications in more specialized journals.

Reviewer #1 (Remarks to the Author):

The authors claim that a Weyl ferromagnetic phase emerges under high pressure. The manuscript appears to be a joint experimental + theoretical paper. The authors present a very interesting claim of an AFM to FM phase transition under pressure and this seems to be the most significant and compelling part of the paper. Predictions of a Weyl metallic state are given on the basis of theory claims.

Reply: We are grateful to Reviewer 1 for confirming the novelty and significance of our work. The Reviewer's valuable comments and suggestions have helped us to further clarify our findings and improve the manuscript in the revised version. We are confident that all the concerns and questions can be addressed.

In my view, all of the items that I have labeled with a "*" should be addressed. I feel that addressing these comments is mandatory requirement before publication.

1) * $\text{La}_2\text{O}_3\text{Fe}_2\text{Se}_2$ is not easy to grow. To my knowledge, the single crystal version has not been reported. Since 2010, expert materials growth research groups have found this material extremely difficult to produce in a single crystal format. Perhaps the authors are aware of a report on the successful growth of the single-crystal form of this material. If so, they should include a citation.

a. *The authors should make it clear/easy for the reader to determine whether the paper is making claims about powder samples or single-crystals. The reader should not have to work hard to determine what sample format is being reported. In my opinion the authors should fix this.

b. *The authors need to explicitly mention how they have managed to grow single-crystals of this material. More explicit growth methods are needed.

c. *If the authors are reporting any results about single-crystal samples, what are the sizes of such single-crystal samples used in each experimental method? Otherwise, it should be made more clear that they are conducting experiments on powders.

d. *A picture of a single-crystal is shown in Fig. 1b. Is this a representative size of the single crystals grown/observed in this material? What is the thickness of a piece such as the one shown in Fig 1b? There is no mention of single crystal refinement.

e. Perhaps I missed this point, but why are single-crystals even mentioned in this paper? Do the authors think that their single-crystals might be used to demonstrate presence of WPs if high-pressure ARPES is performed on these materials?

Reply: We are grateful to Reviewer 1 for these valuable comments and questions. We did not describe our samples clearly in the previous version. First of all, we should point out that our paper is the first one to report the single crystal growth of $\text{La}_2\text{O}_3\text{Fe}_2\text{Se}_2$. The single crystal growth of this material is rather difficult as the reviewer point out, we made big effort to grow the single crystals, however, only tiny single crystals with typical size of 100 μm can be obtained (A representative piece of single crystal is shown in the inset of Fig. 1b, the thickness is about 10 μm). The high-pressure transport measurements were performed on one piece of single crystal with typical size of 50 $\mu\text{m} \times 50 \mu\text{m} \times 10 \mu\text{m}$. We crushed several pieces of single crystals to powder and used for the high-pressure XRD experiments. We added details of the sample growth and experiments in the revised manuscript. We performed single crystal XRD as shown in Fig. 1b, the single crystal rocking curve is shown in Supplementary Fig. 1a, the FWHM is about 0.16 degree, indicating high quality

of our single crystal.

We used single crystals for our high-pressure experiments due to the following reasons:

1. The powder samples of $\text{La}_2\text{O}_3\text{Fe}_2\text{Se}_2$ usually contain impurity phases, and we cannot make sure whether the loaded sample for the high-pressure experiments is pure.
2. For the transport measurements, the boundary scattering of the powder sample may play a critical role, which will blur the intrinsic transport properties.
3. Anisotropy transport properties can be measured on the single crystal.

2) *Regarding the powder diffraction data, there are some 0 GPa peaks that do not appear at higher pressure. The authors should address this.

Reply: The XRD peaks broaden at high pressure due to the pressure inhomogeneity which is inevitable in the high-pressure experiments. For this reason, some ambient-pressure peaks disappear at high pressure, which originate from either overlapping with the other peaks or the low intensity induced by peak broadening. All the peaks shift to the higher angle due to the shrinkage of the lattice parameters, thus some peaks cannot be detected due to the limited angle range of our high-pressure XRD measurements.

3) *The authors need to give an indication of what the sample quality is for the reported materials. More information on the quality of the Rietveld fit should be given; what are some figures of merit for the fitting. For example, can the authors provide details on the Rietveld fitting quality parameters, and an indication of their crystals' quality compared to published crystals?

Reply: We provided the Rocking curve for our as-grown $\text{La}_2\text{O}_3\text{Fe}_2\text{Se}_2$ single crystal as shown in Supplementary Fig. 1a. The FWHM is about 0.16 degree, indicating high quality of our single crystal. We also performed the Rietveld fitting of our powder XRD at 0 GPa in the diamond anvil cell as shown in Supplementary Fig. 1b. Although the R_{wp} and χ^2 are relatively large due to the limited angle range of our high-pressure XRD measurements, the fitting results of our samples are consistent with previous results as shown in Supplementary Fig. 1c.

4) *Some details of the diamond anvil cell should be given in the Methods section. If there is a instrumentation paper written about the diamond anvil cell, the authors should cite.

Reply: We are grateful for Review 1's kind suggestions. We added more details of our diamond anvil cell and cited the related paper in the revised manuscript.

5) *The statement "This implies the existence of six pairs of WPs..." is insufficient. For an article in a general journal, the authors need to provide more explanation of the general readership.

Reply: Thank Reviewer 1 for his/her kind suggestions. We admit that we did not clearly show the presence of six pairs of WPs in the original paper. In one of coauthors' previous papers [Ref. [36], i.e., *Phys. Rev. Lett.* 122, 057205 (2019)], the only use of parity arguments by first-principle calculations is a powerful recipe for identifying ferromagnetic (FM) phase with odd or even pairs of WPs. In this work, our first-principles calculations demonstrated that there are six pairs of WPs in the FM phase of $\text{La}_2\text{O}_3\text{Fe}_2\text{Se}_2$. To clearly understand the behind mechanism, we have done an analysis of parity eigenvalues of the occupied Bloch states at the eight TRIM points. The results show that there are two parallel time-reversal invariant planes (i.e., $k_2=0$ and $k_2=0.5$) with nonzero Chern number $C=1$ [see Supplementary Fig. 6]. In this case, the gapless states in a FM metal require

the presence of 2D planes with $C = 0|_{k_2=k_1}$ ($0 < k_1 < 0.5$). The topological analysis satisfies the FM metallic phase of $\text{La}_2\text{O}_3\text{Fe}_2\text{Se}_2$, in which there are two 2D planes with $C = 0|_{k_2=k_1}$ ($0 < k_1 < 0.5$), resulting in six pairs of Weyl points are present in the whole Brillouin zone (see detailed arguments in Ref. [36]).

Based on Reviewer 1's comments, we have also updated the related statement to clarify this issue in the general readership.

6) *I believe the most important open issue to be understood in this paper is the antiferro-to-ferromagnetic phase transition upon application of external pressure.

Reply: We thank Reviewer 1 for bringing up this valuable comment. In our work, we understood that the transition of magnetic ground state from AFM Mott insulating phase to FM metallic phase is dominated by the spin interactions of magnetic atoms, i.e., the nearest-neighbor interaction of Fe-Fe contribution J_1 , the next-nearest-neighbor interaction of Fe-Se-Fe contribution J_2 , and the next-nearest-neighbor interaction of Fe-O-Fe contribution J_2' , which all exhibit a sudden change around 40 GPa [see Fig. 4b]. The evolution of magnetic interactions as a function of pressure agrees with our resistivity measurement, and such phenomenon may attribute that applying external pressure enlarges bandwidth and diminishing of electron correlation. However, we also admit that the AFM to FM phase transition upon application of external pressure in the real material can be much more complicated, and thus we made related changes in our revised manuscript to more clearly present this important issue.

7) *The true Mott insulator occurs only in the quantum paramagnetic state. An antiferromagnetic insulator is not really a Mott insulator driven by a competition between hopping and the on-site Coulomb interaction U , but a result of band-folding effects or changes in the unit cell associated with development of antiferromagnetic order.

Reply: We agree with the reviewer's point of view. Actually, $\text{La}_2\text{O}_3\text{Fe}_2\text{Se}_2$ is a well-known Mott insulator. The previous studies indicted that this material can be considered as correlation-induced Mott insulators with the antiferromagnetic (AFM) ground state and shows insulating behavior above the antiferromagnetic transition temperature at ambient pressure (Ref. [25]), confirming that it is a Mott insulator rather than the antiferromagnetic insulator. With the increase of pressure, $\text{La}_2\text{O}_3\text{Fe}_2\text{Se}_2$ also exhibit the insulating feature up to the presence of Mott transition. Our calculated results also reveal that the pressure-driven Mott transition of $\text{La}_2\text{O}_3\text{Fe}_2\text{Se}_2$ is dependent on the on-site Coulomb interaction U [see Fig. 4].

8) *Have the authors taken into account the effect of U in their ab initio calculations. Nothing seems to be mentioned regarding this aspect of the paper. Is DFT+ U being used or something else? This should be better addressed in the paper.

Reply: We are grateful to Reviewer 1 for this insightful question. As pointed out by Reviewer 1, the influence of correlated effects of Fe 3d electrons indeed plays important role on the transition pressure from to AFM Mott insulator state to Weyl FM phase. In our work, the most point is that the pressure-induced magnetic transition from the initial AFM state to FM state can always occur in $\text{La}_2\text{O}_3\text{Fe}_2\text{Se}_2$. The DFT+ U approach were demonstrated as an effective method to study the electronic properties of $\text{La}_2\text{O}_3\text{Fe}_2\text{Se}_2$ (Ref. [19,25]), and thus we have considered the on-site Coulomb interaction U of Fe 3d electrons in our first-principles calculations. The calculated results

have been shown in Fig. 4a (also shown in Fig.R1). Considering that applying external pressure would change the Hubbard correlated energy U , we calculated the energy difference ΔE between AFM phase and FM phase for a series of U values as a function of pressure. The results indicate that the pressure-induced magnetic transition from the initial AFM insulator state to FM metallic state can always occur in $\text{La}_2\text{O}_3\text{Fe}_2\text{Se}_2$, and thus the pressure-modified U values do not affect the main physical results of our work. However, we found that different U values would influence the critical pressure of magnetic transitions. We further confirmed that the critical pressure is around 40 GPa when the value of U is about 2.1 eV, which agrees with our resistivity measurement. This U value is obviously smaller than that in AFM Mott insulator phase in ambient condition [Ref. [25], *Phys. Rev. Lett.* 104, 216405 (2010)], implying that the application of external pressure would enlarge bandwidth and diminishing of electron correlation.

Following the reviewer's suggestion, we have updated corresponding statement to clearly emphasize this issue.

Fig. R1. magnetic phase transition of $\text{La}_2\text{O}_3\text{Fe}_2\text{Se}_2$ polycrystal within different Hubbard U values.

9) *Thus, the "shrinkage of U at high pressure" (line 63) and the emergence of Weyl fermions remains questionable in my opinion. Are the Weyl points discussed in the paper derived in bare band or in the correlated electron cell framework? These items are not very clear. A better elucidation should be given.

Reply: We thank Reviewer 1 for bringing up this important issue. In a FM metallic phase, the correlated effects of transition metal atoms would affect the band structure and even band topology. In our work, we employed DFT+ U calculations with $U=2.1\text{eV}$ to investigate the band topology of $\text{La}_2\text{O}_3\text{Fe}_2\text{Se}_2$ under 48GPa. In this case, all the Weyl points are closed to the Fermi level, and the emergence of Weyl points has been confirmed by the topology analysis and parity arguments. Moreover, we have calculated the band structures of FM metallic phase using other U values, and the results show the band topology has always been preserved. To response, we have updated the related statements to better elucidate this issue.

10) *It is unclear how the authors have calculated " ΔE for a series of U values" (line 146).

Reply: The relative energy ΔE can describe by $\Delta E = 0.25*(E_{\text{AFM}}-E_{\text{FM}})$, in which the ΔE represents the energy difference between AFM state and FM state; E_{AFM} is the total energy of relaxed structure with AFM state; E_{FM} is the total energy of relaxed structure with FM state. When the ΔE is positive

value, the system would belong to the AFM state. Following the reviewer suggestion, we have updated the related description in the revised manuscript.

11) Can the authors clarify the physical meaning of "Significantly, when the value of U is about 2.1 eV, the critical pressure is around 40 GPa, which satisfies our resistivity measurement."?

Reply: Considering that applying external pressure would change the on-site Hubbard correlated energy U , we calculated the energy difference ΔE between AFM phase and FM phase for a series of U values as a function of pressure. The results indicate that the pressure-induced magnetic transition from the initial AFM insulator state to FM metallic state can always occur in $\text{La}_2\text{O}_3\text{Fe}_2\text{Se}_2$, and thus the pressure-modified U values do not affect the main physical results of our work. However, we found that different U values would influence the critical pressure of magnetic transitions. We further confirmed that the critical pressure is around 40 GPa when the value of U is about 2.1 eV, which agrees with our resistivity measurement. Based on Reviewer 1's questions, we updated this sentence in the revised version.

12) *Concerning the resistivity measurements, the authors should clarify the origin of the low-temperature resistivity upturn clearly seen in the "metallic phase" at pressure up to 71.3 GPa in Fig. 1d. Is this related to Kondo physics or not? (For details regarding the changes in resistivity due to Kondo effect, see PRB 98, 035114 (2018) and references therein.)

Reply: We thank the Reviewer 1 for point out this issue. We agree with the reviewer that the low-temperature upturn of the resistivity may be related to the Kondo effect. The low-temperature resistivity shows a logarithmic increase and gradually becomes nearly saturated towards low temperatures (as shown in Supplementary Fig. 2, also shown below for reviewer's convenience), which is expected for the Kondo effect. We added the related description and references in the revised manuscript.

Fig. R2. Enlarged area of the temperature dependence of resistivity for $\text{La}_2\text{O}_3\text{Fe}_2\text{Se}_2$ single crystal.

Note that the low-temperature resistivity shows weak upturn when the material becomes metallic. The low-temperature resistivity shows a logarithmic increase and gradually becomes nearly saturated towards low temperatures.

Reviewer #2 (Remarks to the Author):

The authors report a pressure-induced transition from a Mott insulator state to a ferromagnetic Weyl metal state in the iron oxychalcogenide $\text{La}_2\text{O}_3\text{Fe}_2\text{Se}_2$, by combining high-pressure magneto-transport measurements, XRD measurements, and first-principle calculations. I have several questions to the authors as described below;

Reply: We thank the reviewer for carefully reading the manuscript and valuable comments. As following, we would be point-to-point to reply the reviewer's questions.

1. Regarding the resistivity curve under pressure shown in Fig. 1d, why is the kink structure corresponding to the magnetic transition temperature so broad? Normally, ferromagnetic transition may appear as a clearer kink structure in resistivity. Furthermore, why is the resistivity enhanced at lower temperature (e.g. below ~ 30 K for 37GPa)? Does this mean that the system is still an insulator at the lowest temperature?

Reply: We are grateful to Reviewer 2 for these valuable comments and questions. The broad kink of the resistivity related to the ferromagnetic transition is possibly due to the strong FM fluctuation around T_c , which would significantly affect the resistivity and blur the transition around T_c . Actually, in some FM metals, such as Fe_3GeTe_2 thin flake (*Nature* 563 (2018) 94-99), the T_c is difficult to be extracted from the resistivity curves. The low-temperature resistivity upturn is possibly due to the Kondo effect, since the low-temperature resistivity shows a logarithmic increase and gradually becomes nearly saturated towards low temperatures (as shown in Supplementary Fig. 2, also shown in Fig. R2). We updated the related description in the revised manuscript.

2. About the Hall resistivity shown in Fig. 2b&c, the sharpness of the hysteresis curve is clearly different below 41.8 GPa and above 45.7 GPa. Also, the coercive field seems to take the minimum around 50 GPa. What are the possible origins of these observations?

Reply: We thank Reviewer 2 for pointing out this problem. The sharpness of the hysteresis curve and the coercive field can be related to the change of the FM grain size and interaction between different FM grains at high pressure. We added the related description in the revised manuscript.

Although the data quality is amazingly high given the very high pressure they have reached, I'm afraid that this work may not attract a broad interest from the readers. One reason is that I cannot see any direct relationship between the Mott physics at the ambient pressure and the band topology at the high-pressure.

Reply: In our work, we reported the tuning of the ground state from the AFM Mott insulator to the Weyl FM metal by pressure. The AFM ground state and paramagnetic state of $\text{La}_2\text{O}_3\text{Fe}_2\text{Se}_2$ corresponds to \mathbb{Z}_2 classification, while the FM state after Mott transition corresponds to \mathbb{Z} classification. Then, we use parity analysis to verify the evolution of band topology with the increase

of pressure. Though calculating the inversion eigenvalues of occupied bands at the eight TRIM points, we found that the AFM Mott insulating phase of $\text{La}_2\text{O}_3\text{Fe}_2\text{Se}_2$ as well as the quantum paramagnetic state at the ambient pressure both are trivial insulators. On the contrary, in the FM metallic phase at high-pressure, the product of the inversion eigenvalues of the occupied bands at the eight TRIM points is 1 and two paralleled surfaces with the same Chern number $C=1$ are present, indicating the emergence of nontrivial band topology [Ref. [36], i.e., *Phys. Rev. Lett.* 122, 057205 (2019)]. As is well-known, the Mott physics is due to the large U , while U can be greatly suppressed at high pressure, thus leading to the insulator-metal transition accompanied by topological transition. Therefore, we can accordingly conclude that the pressure-driven topological Mott transition occurs in $\text{La}_2\text{O}_3\text{Fe}_2\text{Se}_2$. As far as we known, it is the first time to demonstrate high pressure as an effective method to tune the ground state in the strongly correlated materials and search for the exotic topological quantum states. More importantly, these findings may bridge the gap between band topology and Mott insulating states. We believe our work can attract a broad interest from the readers.

It is highly natural that a Mott insulator finally becomes a metal under pressure, and the appearance of the Weyl points seems to simply originate from its crystal structure.

Reply: We are grateful to Reviewer 2 for providing this comment. Here, we agree with the review that pressure induced insulator-metal transition is already observed in several Mott insulators. However, the pressure-driven transition from the Mott insulator to FM Weyl metal is the first time reported as far as we known. In addition, we respectfully disagree with the claim “*the appearance of the Weyl points seems to simply originate from its crystal structure*”. Our experiment did not show any structural phase transition with increasing pressure. Thus, the AFM Mott insulating phase and FM Weyl metallic phase have the same crystal structure, but these two phases possess distinct band topology. We indeed used symmetry arguments to investigate the magnetization-dependent Weyl points, which is the common approach to discuss the symmetry-protected topological physics. The symmetry of crystal structure is only the necessary condition for the presence of band topology, and the FM Weyl physics of $\text{La}_2\text{O}_3\text{Fe}_2\text{Se}_2$ at high pressure results from the interplay of crystal structure, electron correlations, and magnetic orders.

Another point is that there is no experimental data which is directly related to the presence of Weyl points (e.g. “giant” anomalous Hall effect or chiral anomaly often reported in Weyl semimetals). For these reasons, I recommend its publications in more specialized journals.

Reply: As the Reviewer 2 mentioned, limited by the high-pressure experimental technique, we do not have direct evidence of the Weyl points in the experiments. Fortunately, we observed that the measured AHE is at the same order with the calculated intrinsic AHE, and the sign change of the AHE can also well explained by the slightly change of the Fermi energy at high pressure, these discoveries could support the existence of Weyl points.

REVIEWERS' COMMENTS

Reviewer #1 (Remarks to the Author):

I have reviewed the edits that have been made by the authors. The majority of the sample description edits have been made. In addition, the authors had modified the manuscript in an effort to address nearly all the other points that we initially mentioned.

The claim of the observation of a the pressure-driven topological Mott transition should be of interest to the readership. This stands as a significant claim of the manuscript.

In my view, the paper can move forward to publication.

Reviewer #2 (Remarks to the Author):

I'm satisfied with the point-by-point response from the authors. I believe that this work is now worth publishing in Nature Communications.